# Soccer academy practitioners' perceptions and application of bio-banding

**Chris Towlson**[1]*, **Demi Jo Watson**[1], **Sean Cumming**[2], **Jamie Salter**[3], **John Toner**[1]

**1** School of Sport and Exercise Science, University of Birmingham, Birmingham, United Kingdom,
**2** Department for Health, University of Bath, Bath, United Kingdom, **3** School of Sport, York St John University, York, United Kingdom

* c.towlson@hull.ac.uk

## Abstract

The primary aims of this study were to examine the application of maturity status bio-banding within professional soccer academy programmes and understand the methods employed, the intended objectives, and the potential barriers to bio-banding. Using a mixed method design, twenty-five professional soccer academy practitioners completed an online survey designed to examine their perceptions of the influence of maturation on practice, their perceptions and application of bio-banding, and the perceived barriers to the implementation of this method. Frequency and percentages of responses for individual items were calculated. In the next phase of the study, seven participants who had experience with, or knowledge of, the bio-banding process within an academy youth soccer setting were recruited to complete a semi-structured interview. Interview data was transcribed and analysed using a combination of deductive and inductive approaches to identify key themes. The main findings across the two phases of the study were that [1] there is consensus among the practitioners that the individual effect of maturation impacts their ability to accurately assess the soccer competencies, [2] the majority (80%) of the sample had implemented bio-banding, with practitioners showing a clear preference for using the Khamis and Roche method to bio-band players, with the greatest perceived benefit being during maturity-matched formats, specifically for late or post-PHV players, [3] Practitioners perceived that bio-banding enhances their ability to assess academy soccer players, and [4] practitioners who have used bio-banding believe that the method is an effective way of enhancing the perception of challenge thereby providing a number of psycho-social benefits. Findings suggest that a collaborative and multi-disciplinary approach is required to enhance the likelihood of bio-banding being successfully implemented within the typical training schedules across the adolescent phase of the player development pathway.

## Introduction

The confounding influence of maturation on youth soccer player (anthropometric, physical, technical, tactical, and psycho-social) characteristics which are considered important by practitioners [1, 2] during talent identification and athletic processes are well-documented [3–7].

**Data Availability Statement:** All interview data and question files are available from the University of Hull open access database: https://hull-repository.worktribe.com/output/4024774.

**Funding:** The author(s) received no specific funding for this work.

**Competing interests:** The authors have no conflicts of interest to declare.

The timing of maturation is highly individualised, independent of decimal age [8, 9], and is often characterised by male academy soccer players experiencing heightened rates of anthropometric (e.g., stature,) development spanning the ages 9.7 to 15.2 years [8, 9]. The peak of accelerated growth in stature is typically referred to as peak height velocity (PHV) [10–13], and is often used as a growth landmark in which to assess maturity timing of adolescent soccer players [6, 14–17]. Using PHV as an identifiable reference point to assess player maturity timing is of relevance and importance to academy practitioners, given that players who undergo PHV earlier (i.e., achieve PHV earlier than peers) are often characterised as possessing temporary, enhanced stature, body-mass and key physical fitness (i.e., speed, power) characteristics are typically over-represented within academy programmes [3–5]. Such selection bias are often to the detriment of later maturing players, who may well be equally technically talented, but are less likely afforded a equitable playing environment in which to demonstrate their ability. The premature de-selection of later maturing players (i.e., achieve PHV later than peers) from development pathways will likely homogenise the selection pool with players perceived suitable to be successful at time of selection and/or play key roles where aerial and power advantages are considered preferable (i.e., defence). This limits the variety of players within the selection pool in which senior soccer teams can select from.

Criterion [18] measures for identifying maturation status are rarely available within applied sporting environments. Therefore, maturity offset (i.e., age at PHV–decimal age), age at PHV [10–12] or percentage of final adult height (PAH) [19] methods are often used to estimate maturation status [20]. Bio-banding is considered to be a method to exclusively [21] reduce maturation-selection within team sport [22, 23]. Bio-banding is the re-categorisation of adolescent athletes from chronological aged-ordered groupings into maturation specific groups (i.e., pre-PHV, circa-PHV or post-PHV or thresholds of percentage adult height [24–31]) with the assumed objective to reduce the large within-group variations brought about by individual variations in the timing of PHV across a specific age group. Using popular methods to estimate maturation status [32], bio-banding has been shown to be an effective method for creating equitable playing environments by reducing within group, maturity-related variations in stature and body-mass [24]. The application of maturity-matched (pre-PHV vs pre-PHV, circa-PHV vs circa-PHV, post-PHV vs post-PHV) bio-banding within academy soccer programmes can potentially remove maturity-related differences in locomotor match-activity profiles, whilst manipulating technical and tactical player loads [27, 30, 31, 33]. Maturity-matched bio-banding has also been shown to be well-received by players and key-stake holders [28, 29, 34], with players situated at either end of the maturation continuum stating that they influence match-play more when competing against others of similar maturation [28, 29]. That said, it has also been demonstrated that maturity miss-matched (pre-PHV vs post-PHV, pre-PHV vs circa-PHV, post-PHV vs circa-PHV) bio-banded formats may afford players (particularly later maturing players) the opportunity to demonstrate important psychological characteristics [1, 2], which are perhaps more readily identifiable during difficult playing environments when players are faced with adversity or problematic scenarios which are experienced during mixed-maturity match formats [26]. However, despite clear evidence to suggest that bio-banding is being embraced within academy soccer programmes [26, 28–31, 35], it is currently unclear to what extent bio-banding is being used, how it is being used, and for what purpose. In addition, for bio-banding to be applied more broadly across academy soccer programmes, it is also important for policy makers and prospective research designers to understand the situational and environmental factors which practitioners are confronted with when attempting to apply bio-banding match formats. Therefore, using a mixed-method approach via use of a cross-sectional survey (part A) and interview (part B) design, the aims of the present study were to examine the application of maturity status bio-banding within professional soccer academy

programmes and understand the methods employed, the intended objectives, and the potential barriers to bio-banding.

## Methods

### Part A–Cross-sectional survey design

**Participants.** Having institutional ethical consent (FHS302) and using convenience sampling, twenty-five professional soccer academy practitioners gave their informed consent to participate the survey. Practitioners working within Elite Player Performance Plan (EPPP) [36] academy development programmes (category 1: n = 14 (56%); category 2: n = 4 (16%); category 3: n = 7 (28%)) completed an online survey (https://www.onlinesurveys.ac.uk/) taking approximately 10 minutes to complete. As with previous survey designs [2, 32, 37], practitioners were required to adhere to in/exclusion criteria questions including: Have you previously completed (and submitted) responses to this survey?; Are you 18 years old or above?; Confirm that you have read the participation information sheet?; Are you currently working within EPPP affiliated club?; Have you already completed this survey?; Failure to respond, or adhere to these questions resulted in the practitioners being prevented from completing the survey and being redirected to a thank you page. Responding practitioners comprised of technical coaches (n = 5, (20%)), sport scientists (n = 12, (48%)), performance analysts (n = 1, (4%)), strength and conditioning coaches (n = 3, (12%)), injury specialists (n = 1, (4%)) and practitioners with non-specific senior management roles (n = 3, (12%)), who fulfilled positions within Foundation (U9 to U11: n = 2, (8%)), Youth (U12 to U16: n = 10, (40%)), and Professional (U17 to U21: n = 7, (52%)) development phases of the EPPP. Practitioners had been in post for on average 22.8 months and 24 (96%) respondents were employed full-time by an academy. To capture practitioners normal working practices, the survey was electronically distributed to practitioners during the first trimester (August to October) of the 2021–2022 English domestic soccer season (August to June). An invitation to complete the survey was sent via email to prospective respondents, with a follow up email one month later. Furthermore, professional soccer clubs were invited to distribute the survey internally to appropriate staff and staff were only permitted to complete and submit survey responses once. In addition, an invite accompanied with a link to complete the survey was also distributed via social media. The content validity of the survey was addressed via discussion with suitable academic staff (n = 3) who all possessed a relevant PhD and full-time academy soccer practitioners (n = 3) working within category 1 and 2 EPPP academy systems.

**Survey content.** The survey contained 38 questions across four individual sections (Section 1: "General information"; Section 2–4 "Perceived influence of maturation on practice"; Section 3: "Perceptions of bio-banding"; Section 4 "Multidisciplinary application of bio-banding; Section 5: "Perceived barriers to bio-banding; Section 6: "Conclusion of survey"). All the information disclosed within section 1 of the survey directly corresponded to the respondent and information stated here was coded and anonymised to ensure respondents could not be identified by the research team. Sections 2–4 examined respondents perceptions regarding the influence maturation status has on their ability to assess anthropometrical, physical, technical, tactical, and psycho-social player characteristics and their perceptions and application of bio-banding. Responses here were given either using multi-choice or five-point Likert scale questions with qualitative anchors to best suit the narrative of the question being asked.

### Section 1: General information

Section 1 was comprised of 7 multiple-choice questions which were chosen and designed to establish a broader context about the respondent which may offer plausible explanation for

certain responses. The required information within this section included the EPPP category (i.e., category 1, 2, 3 or 4) of the academy they work within, the primary phase (i.e. Foundation Development Phase (U9 to U11), Youth Development Phase (U12-U16) or Professional Development Phase (U17 to U23)) of the EPPP they work within, nature of their employment (i.e. Full-time, part-time etc), primary role (coach, performance analysts, sport scientist etc), and duration within current position.

## Sections 2 to 4: Perceived influence of maturity status and application of bio-banding

Respondents were required to answer using either a multi-choice (multiple answers were permitted), five-point or six-point Likert scale. A six-point Likert scale was used for questions where it was considered appropriate to provide a "I don't know option" when respondent knowledge was being assessed rather than agreement being sought per se. Agreement with statements relating to the application and perceptions of bio-banding was established using a five-point Likert scale (e.g., "I feel bio-banding enhances the assessment of physical characteristics in academy soccer players"–Strongly disagree, Disagree, Neutral, Agree, Strongly Agree). Structure of the questioning was repeated throughout the survey whereby respondents were asked questions relating to their perception of bio-banding as a format to assess/control for anthropometrical, physical, technical, tactical, and psycho-social player characteristics. These broad groups were chosen on the basis that one of the key objectives of the EPPP is to develop more and better 'home grown' players who are eligible for international representation and the widely used Football Association, Four (Technical/Tactical, Psychological, Physical and Sociological) Corner Model for long-term player development [38] was considered an appropriate framework to base survey questions on.

## Section 5: Perceived barriers to bio-banding

The information in this section was gathered using a five-point Likert scale (e.g., What do you feel are the contributing factors to why you have not used bio-banding?"–Very low factor, Low factor, Moderate factor, High factor, Very high factor). Participants were also asked: "do you feel bio-banding is of greater benefit for early, on-time or late maturing athletes?"–No benefit, Minimal benefit, Neutral, Some benefit, Greatest benefit. In order to ensure all avenues were explored, an additional multiple-choice question was included–"Do you feel there are any other barriers to bio-banding which are not stated above?"–Yes (please specify additional barriers below), No.

## Section 6: Conclusion of survey

The final section included three questions which were designed to allow participants to summarise their overall viewpoints. Two questions used a five-point Likert scale, such as "I feel bio-banding enhances the assessment of physical characteristics in academy soccer players"– Strongly disagree, Disagree, Neutral, Agree, Strongly Agree as well as "Having completed this survey how likely are you to use bio-banding"–Highly unlikely, Unlikely, Unsure, Likely, Highly Likely. The final question sought general consensus on when practitioners believe players should be introduced to bio-banding: "From which stage of development do you believe players should be introduced to bio-banding?".

### Part B–Interview

**Participants.**    Survey respondents who indicated that they were willing to participate in an interview were contacted via email and informed of the purpose of the interview and what it

would entail. Participants were also recruited through the researchers' personal and professional networks. All participants were required to have had experience with, or knowledge of, the bio-banding process within an elite youth soccer setting. The sample consisted of 7 male practitioners currently working within a UK professional soccer academy (National FA academy system: n = 1; English Premier League: n = 3; Scottish Premier League: n = 1; Championship: n = 1; League 2: n = 1) as either a sport scientist (n = 3), strength and conditioning coach (n = 2), academy manager (n = 1) and performance manager (n = 1). All participants signed consent forms prior to the interviews.

**Interview process.**   The study is couched within a post-positivist paradigm which has a number of implications for how the interview data was collected, analysed and represented. For example, interviews were informed by survey data, existing literature and standardised across participants. A single interview was used, and data was subjected to a combination of deductive and inductive analysis. Peer debriefing was conducted to enhance the trustworthiness of the findings and data were represented using a realist form characterized by experiential authority and the participant's point of view.

The interview schedule was informed by common themes within the survey data which were independently identified by three of research team, accompanied by appraisal of existing literature on bio-banding, and discussions with expert collaborators. Interviews were semi-structured in nature which meant that a pre-planned interview guide (see associated supplementary open access repository) was used to ask participants a series of focused and open-ended questions about their knowledge and perceptions of bio-banding. The semi-structured nature of the interviews meant that the order and phrasing of these questions were adapted according to the flow of the conversation. The interview guide sought to address a wide range of questions including participants rationale for using bio-banding, their perceptions of its use and the perceived barriers for its implementation. Focused questions were used to follow up on survey responses such as "why do you feel players should be introduced to bio-banding from the development phase as highlighted in your survey response?". Open-ended questions such as "Please tell me about your experiences using bio-banding" were used to elicit rich descriptions of experience. Follow-up probes or curiosity-drive questions (e.g., "Can you elaborate or explain in more detail why you think maturity matched bio-banding might prove beneficial to athletic learning and development?") were used to encourage more elaborate and in-depth responses. Some probes were detail oriented whilst others sought to encourage elaboration (e.g., "Can you give me an example of a player who benefited from maturity matched bio-banding?") or clarification (e.g., "I'm not sure I understand what you mean by the term "YDP" (i.e., Youth Development Phase). Can you help me understand what that means?"). The use of prompts was accompanied by active listening which involves being attentive and responsive to the interviewee [39]. Examples of active listening included restating the interviewee's message and responding empathetically and these approaches can help increase the length and depth of responses. Interviews concluded by asking participants whether they'd like to add anything that hadn't been covered during the course of the interview. Interviews were conducted and recorded via Microsoft teams' meetings. An automated transcript was produced by Microsoft teams and the researcher subsequently listened back to interviews to ensure that the transcription was accurate. Interviews lasted an average of 43 minutes.

## Data analysis

**Statistical analysis.**   For part A (cross-sectional survey design), raw data were exported to Microsoft Excel (Microsoft Corp, Redmond, WA) and the frequency and percentages of responses for individual items were calculated. To establish a binary (agree or disagree; high

factor or low factor) interpretation, data for composite component data points (i.e., strongly agree-agree, strongly disagree-disagree; very high-high; very-low;) were aggregated. Similar to our previous research [40], and given that the present study collected limited, non-parametric categorical data, it was considered to be unviable to statistically analyse data due to the violation in some statistical assumptions (i.e., Pearson's chi-square test requires expected frequencies to be >5 in ≥80% and ≥1 in 100% of cells or categories) [41]. Therefore, given the cross-sectional and descriptive objectives of this study, data were reported in a descriptive manner.

For part B, following transcription (see associated supplementary open access repository), data were analysed using a directed approach to content analysis [42]. First, transcribed interviews were read several times to gain a clear comprehension of the participants' responses and then subjected to line-by-line analysis. Sentences from the interview transcripts were segmented into phrases that encompassed the participants' perceptions regarding the utility of bio-banding. Next, a combination of inductive and deductive approaches was used to identify meaning units which were subsequently grouped together to form emergent categories (lower-order themes) based on their similarity to each other and distinction from other categories [43]. The deductive element of this process involved coding segmented text using existing theory/predetermined codes. The inductive element involved assigning a new code to any text that could not be categorised using the initial coding scheme [42]. This process was then repeated to generate higher-order themes. Two techniques were employed to enhance the trustworthiness of our data. First, peer-debriefing took place during the data analysis process and these involved members of the research team challenging the primary researcher's initial interpretations of the data [44]. This process sought to establish a general agreement amongst the research team as to how the data was been coded. Two of the researchers identified themes independently and then acted as 'critical friends' by questioning each other's interpretations. Secondly, trustworthiness was enhanced by requesting a third researcher to cast a critical eye over the results and to encourage the team to consider alternative readings of the data.

## Results

### Part A—Cross-sectional survey design

Practitioner aggregated (i.e., sum of strongly disagree and disagree; sum of strongly agree and agree) responses suggest they agreed on the individual effect of maturation on academy soccer players' physical (96%), technical (60%), tactical (48%) and psycho-social characteristics (48%) (Table 1). A similar trend was also reported for practitioners perceived agreement on that individual maturity-related differences can also confound their ability to accurately assess the physical (68%), technical (44%) and technical (56%) competence of a child (Table 1). Practitioners' data showed that they tended to assess maturation either monthly or quarterly across the Foundation (monthly (25%), quarterly (50%)), Youth (monthly (40%), quarterly (50%)) and Professional (monthly (15%), quarterly (50%)) Development Phases of the EPPP. With 10% and 20% of the Foundation and Professional Development Phases respectively stating that they do not assess maturation within these EPPP phases.

Response data showed that 80% of the participating practitioners had implemented bio-banding, with 80% of these practitioners using the Khamis and Roche [19] method (i.e. PAH) to bio-band players. Maturity offset (Fransen, Bush [10] (5%), Moore, McKay [12] (5%), Mirwald, Baxter-Jones [11] (5%)) and skeletal maturation (5%) methods were also shown to have been implemented by practitioners. Responses showed that practitioners felt that bio-banding was of greater benefit for early or post-PHV (80%) and late or post-PHV (92%) players. With less certainty being shown for on-time or circa-PHV (48%) players. A near even distribution for the application of bio-banding across small-sided games (29%), full match-play (25%), technical training (23%) and strength and

**Table 1. Summary table of practitioner individual and aggregated (i.e., Sum of strongly disagree and disagree; sum of strongly agree and agree) perceived agreement that differences in maturation can impact the development of and their assessment of physical, technical, tactical, and psycho-social characteristics of academy soccer players.**

| Survey question | Strongly disagree | Disagree | Neutral | Agree | Strongly agree | Aggregated disagree | Aggregated agree |
|---|---|---|---|---|---|---|---|
| *To what extent do you agree that differences in maturation status impact the development of. . .* | | | | | | | |
| *Physical characteristics (e.g., power, speed, endurance, strength, agility)?* | 0% | 4% | 0% | 44% | 52% | 4% | 96% |
| *Technical characteristics (e.g., passing, shooting, ball control, ability to use both feet, touch)?* | 0% | 12% | 28% | 40% | 20% | 12% | 60% |
| *Tactical characteristics (e.g., pitch exploration, team tactics, opponent, counterattack)?* | 8% | 12% | 32% | 44% | 4% | 20% | 48% |
| *Psycho-social characteristics (e.g., creativity, attitude, resilience, confidence)?* | 0% | 8% | 20% | 40% | 32% | 20% | 48% |
| *To what extent do you agree that maturity-related differences impact your ability to accurately assess the. . .. competence of a child?* | | | | | | | |
| *Physical development characteristics (e.g., power, speed, endurance, strength, agility) impact your ability to accurately assess the physical competence of a child.* | 4% | 16% | 12% | 36% | 32% | 20% | 68% |
| *Technical development characteristics (e.g., passing, shooting, ball control, ability to use both feet, touch) impact your ability to accurately assess the technical competence of a child.* | 4% | 20% | 32% | 28% | 16% | 24% | 44% |
| *Tactical development characteristics (e.g., pitch exploration, team tactics, opponent, counterattack) impact your ability to accurately assess the tactical competence of a child.* | 8% | 24% | 32% | 28% | 8% | 32% | 36% |
| *Psychological development characteristics (e.g., creativity, attitude, resilience, confidence) impact your ability to accurately assess the psycho-social competence of a child.* | 8% | 12% | 24% | 44% | 12% | 20% | 56% |

conditioning sessions (21%) was evident. However, only 2% of respondents used bio-banding for psycho-social specific sessions. Practitioners' responses showed that technical development (22%), physical development (21%) and talent identification were the primary objectives for using bio-banding, with matched bio-banding being a preferred format (33%). Practitioners perceived matched bio-banding to permit enhanced assessment of physical (95%), technical (85%), tactical (60%), and psycho-social (75%) player characteristics in comparison to chronologically categorised player groupings (Table 2). Practitioners' responses also show that maturity-matched bio-banding to be perceived to be the most effective strategy to create an environment to assess players physical (60%), psycho-social (40%) and technical (52%) characteristics (Fig 1). Overall, responding practitioners perceived bio-banding to enhance their ability to assess of physical (80%), technical (68%), tactical (40%) and psycho-social (60%) characteristics of academy soccer players (Fig 2) and that bio-banding should be implemented during the Foundation (16%) and Youth (76%) development phases of the EPPP. With 8% of respondents stating that they feel bio-banding should not be introduced in any phase of the EPPP. Coaches buy-in (36%), lack of bio-banding understanding (32%) and social stigma related to players 'playing down' (32%) were perceived as the highest contributing factors as to why bio-banding may not be implemented within their soccer academy (Table 3). Overall, 15% of respondents stated that either their coaching course and/or academic qualification made them aware of issues related to athlete growth and maturation.

## Results

### Part B–Interviews

Interviews revealed that the majority of the practitioners viewed bio-banding in a positive light and considered the approach to have wide-ranging utility as a talent identification and

**Table 2. Summary table of practitioner individual and aggregated (i.e., sum of strongly disagree and disagree; sum of strongly agree and agree) perceived agreement that matched (e.g., Early vs Early or pre-PHV vs pre-PHV) and miss-matched (e.g., Late vs Early or pre-PHV vs post-PHV) bio-banding permits enhanced assessment of physical, technical, tactical, and psycho-social player characteristics in comparison to chronologically categorised (i.e., U11 etc) player groupings.**

| Survey question | Strongly disagree | Disagree | Neutral | Agree | Strongly agree | I don't know | Aggregated disagree | Aggregated agree |
|---|---|---|---|---|---|---|---|---|
| *Please state your level of agreement for how bio-banding permits an enhanced assessment of the below components of performance when matching (e.g., Early vs Early or pre-PHV vs pre-PHV) players for maturity status in comparison to chronologically categorised (i.e., U11 etc) player groupings?* | | | | | | | | |
| *Physical characteristics (e.g., power, speed, endurance, strength, agility)* | 0% | 5% | 0% | 35% | 60% | 0% | 5% | 95% |
| *Technical characteristics (e.g., passing, shooting, ball control, ability to use both feet, touch)?* | 0% | 5% | 10% | 70% | 15% | 0% | 5% | 85% |
| *Tactical characteristics (e.g., pitch exploration, team tactics, opponent, counterattack)* | 0% | 10% | 30% | 55% | 5% | 0% | 10% | 60% |
| *Psycho-social characteristics (e.g., creativity, attitude, resilience, confidence)?* | 0% | 0% | 25% | 45% | 30% | 0% | 0% | 75% |
| *Please state your level of agreement for how bio-banding permits an enhanced assessment of the below components of performance when pairing (e.g., Late vs Early or pre-PHV vs post-PHV) players for maturity status in comparison to chronologically categorised (i.e., U11 etc) player groupings?* | | | | | | | | |
| *Physical characteristics (e.g., power, speed, endurance, strength, agility)* | 0% | 10% | 25% | 35% | 20% | 10% | 10% | 55% |
| *Technical characteristics (e.g., passing, shooting, ball control, ability to use both feet, touch)?* | 0% | 0% | 25% | 65% | 0% | 10% | 0% | 65% |
| *Tactical characteristics (e.g., pitch exploration, team tactics, opponent, counterattack)* | 0% | 0% | 50% | 40% | 0% | 10% | 0% | 40% |
| *Psycho-social characteristics (e.g., creativity, attitude, resilience, confidence)?* | 0% | 0% | 30% | 50% | 10% | 10% | 0% | 60% |

development tool. It's predominantly used for identification of talented players. Six participants/interviewees had first-hand experience of employing the method. All of these participants used the Khamis and Roche [19] method to assess maturational status. The majority of participants felt that bio-banding was best employed during the youth development phase. Two main themes were identified which portray the perceived benefits of bio-banding and the barriers to its implementation.

**Theme 1: Perceived benefits of bio-banding.** *Introduction of technical and tactical challenge.* Practitioners considered bio-banding to present early and late maturers with a range of different challenges that served to enhance their tactical and technical capacities. 5 of the practitioners thought bio-banding proved especially beneficial for early maturers. To illustrate, Oscar argued that this category of players "need to be playing against players who are at least matched to them, because if they're just playing with late maturers or on time players then they could just rely on the physical side and not so much on the technical or tactical side". For Mason, bio-banding serves an important technical and tactical function for these early maturers:

> If a big player is mixed chronologically, they get the option to just kick the ball around and push people out of the way and that touch doesn't need to be good, it doesn't need to be perfect cause they can take the bad touch and they can just shove someone out the road. But when you match them with people their own size they can no longer do that so you have to force them into a situation where they have to work technically . . . and tactically as

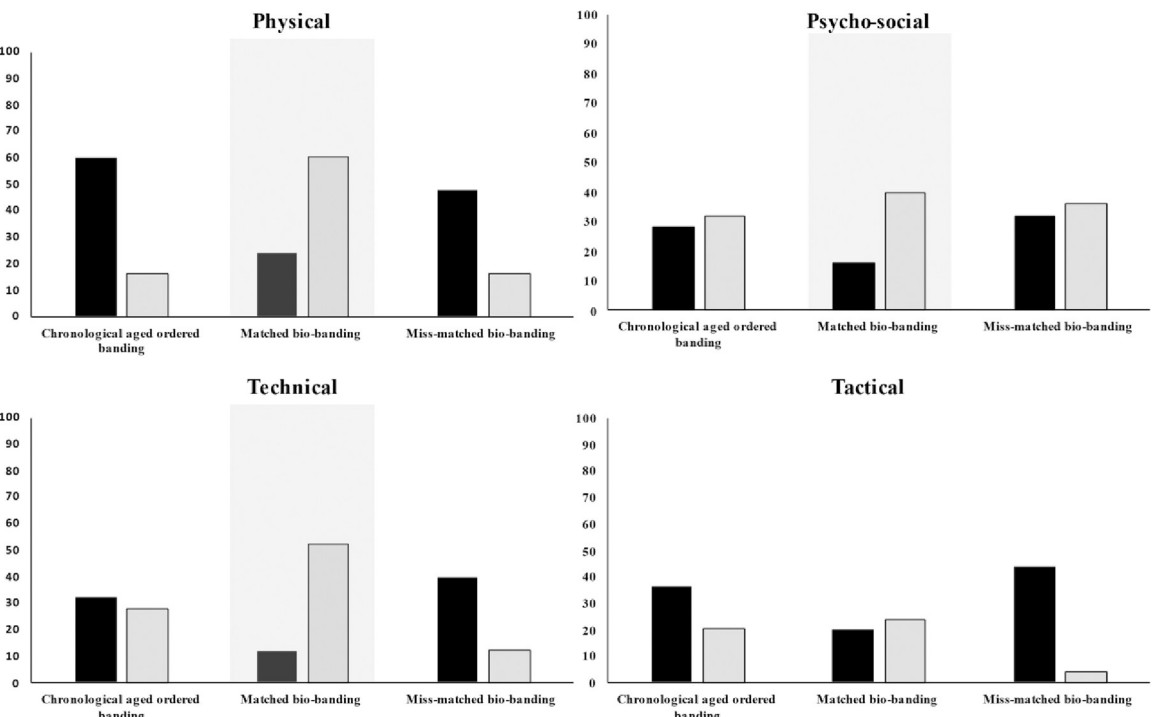

**Fig 1. Percentage distribution of practitioner aggregated (i.e., sum of strongly disagree and disagree; sum of strongly agree and agree) perceived agreement (Black–Disagree; Grey–Agree) of whether they agree chronologically ordered formats (U11, U12, U13 etc), maturity matched (i.e. early vs early, post-PHV vs Post-PHV etc) or maturity miss-matched (i.e. early vs early, pre-PHV vs Post-PHV etc) bio-banding is the most effective strategy to create an environment to assess players physical, psycho-social, technical and tactical characteristics.**

well . . . that's something that sort of ties in because you're trying to teach them to basically problem solve and how to make sense of the chaos within the game . . . so yeah find an environment which forces them to work more technically.

Whilst acknowledging the benefit early maturers might derive from bio-banding, practitioners see bio-banding as equally, if not more important for late maturers. Reflecting on his experiences implementing bio-banding sessions, Mason gave the example of a player

who is playing in an A squad and who for me wouldn't be anywhere near that if we hadn't employed those methods . . . you know he's just constantly playing against people who are too big for him. That's why he's not performing well and being able to strip that back and say when he's actually in a group where the people are sort of physically matched he stands out.

According to Daniel:

the whole point of bio-banding is to create new and challenging opportunities for the players . . . we had an early maturer who (having taken part in bio-banding) then had an increased physical challenge and they then didn't deal with that increased challenge and that was subsequently a developmental point for them moving forward.

Carter sees the introduction of challenge as crucial to the development and cultivation of talent and to prepare these young players:

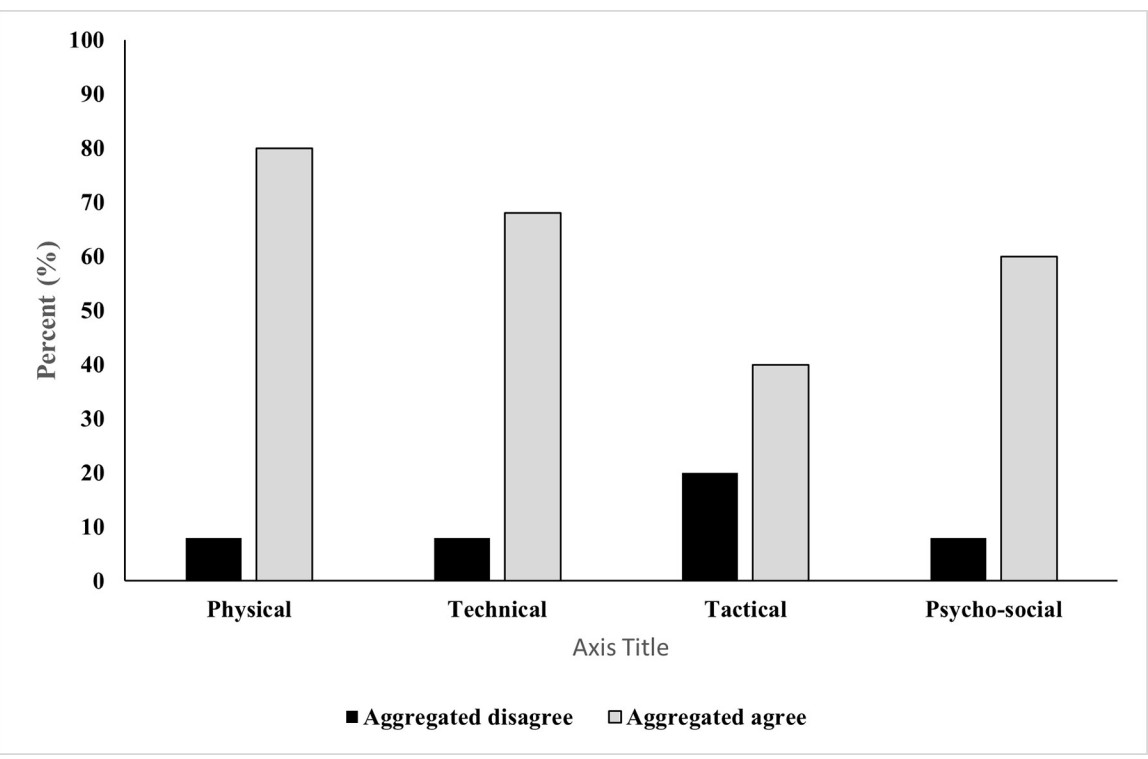

**Fig 2. Percentage distribution of practitioner aggregated (i.e., sum of strongly disagree and disagree; sum of strongly agree and agree) perceived agreement (Black–Disagree; Grey–Agree) of whether they feel bio-banding enhances the assessment of physical, technical, tactical and psycho-social characteristics of academy soccer players.**

**Table 3. Summary table of practitioner individual and aggregated (i.e., sum of strongly disagree and disagree; sum of strongly agree and agree) perception of contributing factors as to why they may not implement bio-banding.**

| Contributing factor | Very high factor | High factor | Moderate factor | Low factor | Very low factor | Aggregated High factor | Aggregated Low factor |
|---|---|---|---|---|---|---|---|
| *Coaches buy-in* | 16% | 20% | 12% | 40% | 12% | 36% | 52% |
| *Lack of bio-banding understanding* | 8% | 24% | 20% | 32% | 16% | 32% | 48% |
| *Social stigma related to players 'playing down'* | 12% | 20% | 12% | 36% | 20% | 32% | 56% |
| *Players buy-in* | 8% | 20% | 16% | 32% | 24% | 28% | 56% |
| *Personnel to implement bio-banding* | 0% | 24% | 24% | 40% | 12% | 24% | 52% |
| *Disruption to the training programme* | 12% | 12% | 28% | 32% | 16% | 24% | 48% |
| *Knowledge of which maturation equation is most appropriate* | 4% | 20% | 24% | 28% | 24% | 24% | 52% |
| *Parents/guardians buy-in* | 8% | 12% | 28% | 28% | 24% | 20% | 52% |
| *Not enough scientific evidence to support use* | 0% | 20% | 20% | 20% | 40% | 20% | 60% |
| *Situational Factors (lack of equipment/resources, players etc)* | 0% | 20% | 24% | 32% | 24% | 20% | 56% |
| *Understanding of how to use maturation equations* | 4% | 12% | 24% | 28% | 32% | 16% | 60% |
| *Players not playing with friends* | 0% | 12% | 16% | 44% | 28% | 12% | 72% |
| *Governing body restrictions of player reallocation legislation* | 0% | 12% | 16% | 36% | 36% | 12% | 72% |
| *Lack of club funds* | 0% | 0% | 8% | 48% | 44% | 0% | 92% |
| *Health and safety* | 0% | 0% | 4% | 36% | 60% | 0% | 96% |

for the senior game . . . it's not always going to be a very sterile environment where everybody is at the same level and stages. You still need to push them on and introduce different challenges . . . to promote that physical challenge for the more developed players is going to be good for them and it's going to allow them to not solely rely on their physical dominance compared to less developed kids.

Carter proceeded to explain that bio-banding might also promote "different training adaptations you would normally get within the regular season (or session), so by slightly manipulating their training will just push them to focus on areas that they might not be focusing on during normal training".

*Psychological challenge*. The bio-banding process was also seen to offer several psycho-social benefits for participants. Aiden points to the social benefits that might be accrued when players are given the opportunity to train with a new group: "It's different, you train in different groups you might get a different coach you might learn something off another player that you wouldn't get from your usual group who you train with every day". Carter felt that bio-banding allows coaches to tease out certain psychological characteristics by identifying those players who might "embrace the challenge of playing against players who are say stronger than them and actually enjoy it or some players may hide away from it". Carter explained that bio-banding might also promote "different training adaptations from what you would normally get within the regular season (or session), so by slightly manipulating their training it will just push them to focus on areas that they might not be focusing on during normal training" as well as "setting expectations and standards for each level". Similarly, Daniel related the case of an athlete who "was moved into a different zone that then provided them with a bit of demotivation about why they were there, but then also provided them a new opportunity to play in a position that they didn't normally play in and that subsequently became that players new position moving forwards". In these cases, bio-banding tests the players resolve by exposing players to new challenges (e.g., playing out of position) Finally, Oscar noted how bio-banding may serve an important psycho-social function by widening the network of athletes they play with:

Bio-banding is a great way of adding in different challenges, not just physically sort psycho-socially as well so if they're grouped together with people who they're not typically used to working with, can they then sort of build great relationships with them? Can they potentially, if they're an older player chronologically, can they go with individuals who are not typically in their age group? Or can they be bit more of a leader whereas they might not be a leader in their actual age group? It's little things like that, which I think can really yeah benefit players going forwards.

Here we see an example of how exposing the players to new and unfamiliar groups might elicit important psychological characteristics such as leadership.

*Reducing injury risk*. A number of the practitioners felt that bio-banding has helped them to identify players who might be at greater risk of growth-related injury as a result of their maturational status. For example, Carter felt that bio-banding "might have allowed us to mitigate or reduce some of the injuries" that they would usually see at the academy. Oscar has employed bio-banding "to highlight when they (the athletes) typically are going to be going through a growth spurt and when there are greater chances of growth-related issues". Assessing/establishing maturational status has allowed Oscar to tailor

injury prevention programmes to their stage of maturation. So, for example, we've got one player at the moment who is in the under 16's who's 92% of his predicted height . . . it's

enabling me to actually produce his gym programs in the evenings to suit his needs a lot more, so working on his hip strengthening and leg strength for the moment just to try and facilitate strength gains and hopefully prevent or reduce the occurrence of injuries

Similarly, Daniel has been assessing maturation "to understand who's going through their growth spurt and who's growing at a rate and subsequently adapted their training program, and we had some real significant decreases in the likelihood of injury for those groups".

**Theme 2: Barriers.**   *Logistics*. Whilst the majority of the participants saw clear value in the role bio-banding might play in the identification and development of talented soccer players, they also outlined a number of barriers that either hinder its implementation or reduce its potential efficacy. The logistical challenge associated with attempting to organise bio-banded sessions represents one of the biggest barriers faced by practitioners. In discussing this challenge, Mason revealed that:

It's basically taken us from March until about 2 weeks ago to arrange like a round robin 6 bio banded games at clubs. So takes months and months and then data collection and then who's going to be available. Who's there, who's not there? I think the challenge to me is organizing and weighing up whether it's actually worth it . . . it's got to make the player better. And I think sometimes the biggest challenge is breaking down what you're doing and asking yourself is this format actually going to make the player better

For Daniel, logistical challenges arise because bio-banding requires:

moving players around between age groups when players all train in their specific age groups but who also might get lifts to sessions in those age groups. There might be particular times for those sessions and then flipping that on its head and moving players around and making them come at different times and potentially not be able to get lifts. And then there's the whole logistical factor that are associated with mixing age groups.

Furthermore, logistical challenges are exacerbated by the difficulty practitioners face in being able to predict with any certainty which players will be available for training. According to Oscar:

So, we work on a week-to-week basis within the academy where the availability of players is literally determined that week. So it's not really something that we can necessarily plan ahead too much. Now, obviously we can have the groupings of all the players with their percentage of particular under height but we can't guarantee that the player is going to turn up the following week. And yeah, I think it's about sort of working around that and having to adapt to it.

A lack of personnel presents sport science staff with yet another logistical challenge: Sean noted:

the personnel obviously is a big one (barrier) for me and I do think personnel are limited with our category . . . bio banding is maybe more of a luxury tool as opposed to something that sort of needs to happen, so we're not able to implement it as much as I would like.

Some of the practitioners found that even if coaches appreciated/recognised the benefits of bio-banding they were reluctant to disrupt their team's preparation for an upcoming competitive fixture by facilitating the use of bio-banded sessions. As Carter put it:

sometimes coaches just want a normal training session with that group. So suddenly if you're preparing for a game that weekend and you're trying to manipulate squads and training groups, that can become difficult. Especially during different stages of the season because some teams are preparing for specific fixtures so you're having to find the balance between preparation and wanting to develop the players physically and technically with all the benefits that come with bio-banding.

While sport scientists might seek to prioritise long-term athletic development, coaches operating within a win-at-all-costs culture may have little choice but to prioritise short-term competitive success. According to Carter, this might mean that "sometimes coaches are resistant to the time that's required to bio-band but not bio banding itself . . . because for some of them they're preparing their team for a game then maybe they're not thinking as much about the long-term development of the players".

*Education/understanding.* A lack of understanding by some coaches as to what bio-banding entails and how it might prove beneficial to athletes is another factor that can hinder its implementation. According to Oscar, this means that "If coaches don't understand it then they potentially will be reluctant to employ it. So, because of that I think it's important to actually educate the coaches on what it is and why we look to employ it". From Sean's perspective: "the battle for moving forward is always going to be making sure that coaches have that discipline to understand the bio-banding process that's in place". Similarly, Daniel noted that "coaches' perceptions and . . . their education around why they're doing it and what the benefits are for each of the individuals, and it needs to be explained that it's not like a holiday camp style of mixing between age groups. It should be a purposeful development tool". However, Daniel acknowledged that even if sports science staff possess a keen understanding of how to implement bio-banding they often "don't have the largest say about what the training schedule and training program looks like and therefore a change in that program to incorporate bio-banding can be difficult". Daniel suggested that one way of addressing this problem might be if sport scientists are "embedded as part of the MPT and involved in joint decision making for the YDP age groups which makes it easier to incorporate bio banding because they have those day-to-day discussions with the coaches". It was clear that many of the participants felt that coach education needed to introduce coaches to bio-banding and explain how it might benefit talent development in academy settings. Oscar stated "I think it's so important to actually educate the coaches on what it is and why we look to sort of employee within the club". The lack of education around bio-banding was also expressed as an issue which potentially needs to be reviewed nationally. Mason considers there to be:

a lack of support, certainly or education. I think the National Association could provide more support and more education in that field and I think we could almost go and help the Clubs understand it, show them how to do it, how to organize it, why it might be beneficial.

All participants who work full-time in clubs felt that coaches' failure to fully understand the rationale for the method was a significant barrier to its implementation. However, the one practitioner who has no experience implementing bio-banding also believes that a lack of education can be detrimental to the process as a whole. To illustrate, Jack felt that "when you're running an Academy program it's really important, I think, to go all in on something or not at all. And not knowing enough about bio banding, to see that as something we go all in on . . . it's just not at my level of knowledge". This statement was corroborated by Aiden who argued that "doing bio banding wrong can really exacerbate the negative side effects or the negative elements of it and therefore can increase the barrier to potentially using new kinds of ideas".

Tellingly, Aiden admitted that he didn't fully understand the method when initially employing it:

> all the smaller players or late maturing players were placed in group order, and same for the bigger and early maturing players for a range of age groups. So, we basically put the later maturing against the early maturing, which I suppose means that we've grouped the right players together so you can see them in their right environment. However, competing against you know the early maturing ones was clearly a massive negative for the later maturing players.

As identified in the questionnaire results, another potential barrier to the use of bio-banding is parental concern about their child being "played down". Again, this might reflect a misunderstanding by parents as to the purpose and rationale for bio-banding. As noted by Mason:

> The big problem I've had when I've worked in academies is that parents become involved and there's external pressure that doesn't need to be there because their 14 year old kid is being told that he's going to play with 13 yearr olds and they think oh my god it's game over, he's being released, he's being played down.

Oscar shared similar concerns when he revealed that: "we don't want to make it look like he is being played down or there's any negative connotations, you want to keep it as open and honest". Mason felt that these concerns may not arise once the rationale for bio-banding is "communicated properly".

## Discussion

The main findings of our mixed-method study are that (1) there is consensus among the sampled practitioners that the individual effect of maturation impacts players physical, technical, tactical and psycho-social characteristics, which likely confounds their ability to accurately assess soccer competencies, (2) the majority (80%) of the sample had implemented bio-banding, with practitioners showing a clear preference for using the Khamis and Roche [19] method to bio-band players, whilst also suggesting no clear preferred format for its application (e.g. match-play, SSG etc), (3) Practitioners perceived that bio-banding enhances their ability to assess the physical, technical and psycho-social characteristics of academy soccer players, and (4) practitioners who have used bio-banding believe that the method is an effective way of reducing player injury risk, enhances the perception of challenge, and provides psycho-social benefits, but successful implementation isn't without situational and logistical challenges.

The over-selection of early maturing, adolescent soccer players who possess (often temporary) enhancements in anthropometric and physical fitness characteristics in favour of their less mature counterparts (who are often characterised as being smaller and having inferior physical characteristics) is well-documented [3–5]. Findings here offer explanation for the possible causes for this selection phenomena, by revealing consensus among practitioners that the individual timing of PHV confounds their ability to identify talented soccer players when using a multi-displinary approach (i.e., physical, technical, tactical, and psycho-social characteristics) for talent identification and development. The implementation of innovative player development frameworks (such as the EPPP [36]) which mandate professional soccer academy programmes to systematically monitor the maturity status of its players has enhanced the utilisation of maturity estimations methods such as percentage estimated final adult height (i.e. Khamis and Roche [19]) and maturity offset (i.e. Fransen, Bush [10], Mirwald, Baxter-Jones [11], Moore, Cumming [45]) [32]. That said, although practitioners identify the associations

between maturity-related changes in anthropometric characteristics and increased risk of injury risk [32], this study confirms previous findings [32] showing some uncertainty exists between practitioners regarding which of the maturity estimation methods they should use to estimate player maturity status. Such inconsistencies are important to identify to ensure a consistent league-wide approach to player maturation monitoring is established to ensure valid longitudinal assessment and tracking. As each individual equation has a level of error inherent within it, with some equations being suggested to possess greater criterion biological maturity validity and reliability than others (see review by Towlson, Salter [6]), such differences might lead to mis-categorisation when implementing popular recategorization methods such as bio-banding [24–27, 30, 33].

The present study showed that 80% of the respondents had implemented bio-banding, with practitioners showing a clear (80%), but not unanimous preference for using the Khamis and Roche [19] method. This obvious preference for the Khamis and Roche [19] method is encouraging given that this method has been shown to possess superior maturity estimation accuracy [46]. Parr, Winwood [46] showed that 96% of a sample of professional academy soccer players experienced PHV during the specified window (85–96% PAH) in comparison to only 61% using the maturity offset approach (± 1-year generic age) [46]. With Salter, Cumming [47] having also showed some disparity in agreement between maturity offset and PAH methods (–1.5 to 1 year). That said, although the Khamis and Roche [19] method has been recognised by researchers and practitioners (including the present study) [6, 46, 47] as being the most suitable equation for estimating the maturity status of academy soccer players, practitioners should be aware that prediction error also manifests within the Khamis and Roche [19] method with median error being reported as 2.4–2.8 cm to 5.5–7.3 cm for those children who are situated on the 50th and 90th normative growth percentiles respectively (see Towlson, Salter [6]).

Although practitioners are recognising the confounding influence of PHV timing on successful talent identification and development, and that those working with academy soccer players are readily implementing bio-banding, results here also suggest that practitioners have no clear preference for its intended application (e.g., match-play, SSG etc). However, practitioners perceive it as having the greatest benefit during maturity-matched bio-banding formats, specifically for late or post-PHV players. Evidence of practitioners actively implementing bio-banding methods is encouraging and suggests that coaches who are responsible for talent identification and development are perhaps taking a research-informed approach to their practices. For instance, maturity-matched bio-banding has been shown to reduce the within group variation of maturity-related anthropometric characteristics (specifically stature and body-mass) [24], whilst also providing an equitable playing environment for players to demonstrate technical-tactical [30, 31, 33], physical [26] and psychological [26] behaviours. However, it is unsurprising that the present study has revealed that a lack of consensus for the intended application of bio-banding exists between practitioners. This is likely due to the fact that although the idea of bio-banding was first considered in relation to child labour laws in the early 1800's (see Malina, Cumming [23]), research on re-categorising adolescent soccer players by maturity status remains in its infancy, with few robust studies [24–27, 30, 33] currently existing which examine the effect of bio-banding on soccer player characteristics. Many of these studies have specifically examined the effect of creating equitable playing environments via the implementation of maturity-matched (i.e., pre-PHV vs pre-PHV or Late vs Late) bio-banding [24, 30, 33], whilst few studies have investigated the effect (positive and negative) of creating worst case playing environments via the use of both maturity matched and miss-matched (i.e. pre-PHV vs post-PHV or Late vs Early) match formats [26, 27]. In general, bio-banding has shown some early evidence of controlling the maturity-related

differences in locomotor characteristics due to the transient, anthropometric, and physical fitness advantages afforded to early maturing (or post-PHV) players' [26, 33]. This is of importance and relevance to talent development and identification practitioners given that technical and tactical player attributes are a key consideration for talent selectors [2] and are influenced by age and advancing maturity in academy soccer [48]. The latter proposition is further supported by evidence which suggests that early maturing soccer players' may attain higher maximal running speeds, cover greater relative high-speed distances, and perform more high-speed accelerations compared to their later maturing counterparts [49, 50]. Such advantages might necessitate the implementation of bio-banding to create an equitable playing environment and negate transient, between player differences likely caused by the individual timings of PHV onset [8, 9]. For instance, the application of maturity-matched bio-banding has shown on-time and later maturing players' tend to perform a greater number of short passes, perform fewer long passes and are afforded greater opportunity to run with the ball during match-play when compared to chronologically age-grouped match formats [33]. This suggests that later maturing players perform a greater number of technical and tactical skills in bio-banded games [29, 30]. In addition to technical actions, the efficacy of bio-banding on locomotor characteristics may also depend on the match format. Lüdin, Donath [30] showed that early maturing players perform a greater number of high intensity acceleration actions during maturity-matched bio-banded matches. Whereas Towlson, MacMaster [26] have shown that few between maturity group differences in locomotor variables exist during the most extreme conditions (i.e. pre-PHV Vs post-PHV players') [26]. Therefore, the effect of maturity status bio-banding on locomotor characteristics are likely dependent on game factors such as relative pitch size, with smaller pitch areas perhaps leading to lower physical strain and increased technical actions [25]. With such findings in mind (including a bias for maturity-matched design bio-banding studies [24, 30, 33]), it is unsurprising that practitioners in this study demonstrated no clear preference for the intended application of bio-bandings (e.g., match-play, SSG etc) and perceive it as having the greatest benefit during maturity-matched formats. That said, although maturity-matched bio-banding affords greater opportunity for later maturing players to engage in leadership behaviours, and provides an opportunity for greater composure on the ball but enhances the sense of pressure to perform well [28, 29], purposely miss-matching players has also shown to have a beneficial effect on important player psycho-social behaviours. For instance, purposefully miss-matching players for maturity status (i.e. pre-PHV vs post-PHV or Late vs Early) has been shown to be an effective strategy to afford players with greater opportunity to display desirable psychological behaviours that are more likely to be displayed during adversity. Towlson, MacMaster [26] have shown that pre-PHV players may receive more positive evaluations for key psychological behaviours (e.g., attitude, confidence, competitiveness, and total psychological score) than post PHV players' during maturity mis-matched bio-banding formats. Such findings may imply that less mature players' are more likely to require superior psychological skills to succeed [51], a suggestion supported by the so-called 'underdog hypothesis' [51, 52]. These findings are of importance to talent development and identification practitioners given that self-regulatory skills often characterise elite athletes from their less-skilled counterparts [53]. Therefore, although responders within the present study exhibited a clear preference for the application of maturity-matched bio-banding methods, practitioners may also wish to consider the use of maturity miss-matched methods depending on the key performance indicator they wish to evaluate.

In addition to talent identification purposes, practitioners who have used bio-banding also perceive that bio-banding is an effective way of reducing player injury risk. This perception is intuitive given that that the timing of PHV aligns with changes in bone density, joint stiffness and subsequent imbalances between strength and flexibility, which contributes to 'skeletal

fragility' [54, 55]. Such maturity-related anatomical adaptations have been suggested to partially explain the phenomenon 'adolescent awkwardness' [6]. This occurs when the trunk and lower-limb length have increased, but soft tissues (e.g., muscle, connective tissue, and nervous system) have not yet adapted to the growing skeletal frame, causing abnormal movement mechanics that negatively impact performance [56, 57]. Such maturity-related adaptations have been associated with increased risk of injury [58]. Recent evidence suggests that a linear relationship between growth rate and injury incidence exists, with higher growth rates being associated with a greater estimated likelihood of injury [59]. With such findings in mind, it is encouraging to note that respondents consider bio-banding to be a viable method to control for, and prescribe maturity specific training loads whilst accounting for skeletal fragility and adolescent awkwardness. However, evidence to support the efficacy of bio-banding to reduce growth-related injury burden is currently absent and qualifying the validity of practitioner perceptions here is difficult.

Despite evidence that suggests both key stakeholders and players recognise the benefits of bio-banding [28, 29, 34], coupled with the emerging support that bio-banding can control for maturity-related differences in anthropometric, physical, psychological, and technical-tactical academy soccer player characteristics [24–27, 30, 31, 33], successful implementation of bio-banding isn't without situational and logistical challenges. Responses in the present study revealed that practitioners perceived logistical challenges such as inter and intra club match organisation, the reallocation of players to fit player training schedules, travel arrangements, unknown player availability and adequate staff availability as being the greatest barriers to implementing bio-banding. Furthermore, findings suggest that some coaches may lack understanding of what bio-banding entails and the benefits it may serve, and this seems to hinder its implementation. Such miss-understanding of bio-banding is unsurprising as we recognise that bio-banding is a relatively novel approach. However, most arguments posited against bio-banding misrepresent or fail to understand its purpose and practice, and the fundamental principles of auxology. Proponents of the 'talent needs trauma' expression [60], for example, describe bio-banding as a 'misplaced solution for the relative age effect (over representation of players being born in the first quarter of the selection year [61, 62])', failing to recognise that maturation and relative age are independent constructs that exist and operate independently (see Towlson, MacMaster [21]). Simply put, bio-banding is not designed to address the relative age effect. Arguments that bio-banding removes the challenge necessary for late maturing players' to optimally succeed also ignore the fact that (i) bio-banding is an adjunct to, not replacement for, age group competition, (ii) late maturing players are generally absent in academy soccer from 14 years, and (iii) the lack of challenge experienced by early maturing players'. In addition to practitioners who do not possess a sport science background with understanding perhaps posing as a barrier to bio-banding, the broader suggestion from the present study is that even if sports science staff possess a research-informed understanding of how and why bio-banding should be implemented, such staff may not possess the decision-making authority to categorise players. Therefore, findings here suggest that encouraging further collaboration between coaching and sports science staff should help permit the embedding of bio-banding as part of typical weekly/monthly training schedules across the adolescent phase of the player development pathway. National governing body provision of coach education opportunities detailing the rationale, purpose, and benefits of bio-banding might further increase their 'buy-in' to this approach.

The objectives of the current study were to examine the application of maturity status bio-banding within professional soccer academy programmes and understand the methods employed, the intended objectives, and the potential barriers to bio-banding. Despite producing a novel insight to bio-banding, we recognise that the sample of responses within this study

are limited. However, we propose that by combining a comprehensive survey within depth interviews, this study has provided a unique and detailed insight in to how a popular and important method to enhance player talent identification and development is being used. Therefore, we feel that this study is the first of its kind to provide insight and first-hand accounts of how bio-banding is being used within professional soccer academies. Importantly for future research design and for prospective users of bio-banding, this study shares the intended uses, challenges, and barriers for implementing bio-banding. In conclusion, there is consensus among the practitioners that the individual effect of maturation impacts players physical, technical, tactical, and psycho-social characteristics, and likely confounds their ability to accurately assess the soccer competencies. In recognising this, the study shows that practitioners are engaging in bio-banding processes and are predominantly using the Khamis and Roche [19] method to bio-band players during match-play activities to mainly assist the evaluation of physical, technical and psycho-social characteristics of late or post-PHV players academy soccer players. For the implementation of bio-banding to succeed, a collaborative approach to its implementation should be taken (i.e., joint decision making across multi-displinary staff) to permit the successful embedding of bio-banding within the typical weekly/ monthly training schedules across the adolescent phase of the player development pathway.

## Acknowledgments

The authors would like to acknowledge and thank the contributions of the survey respondents and interviewees. Without these, this study would not have been possible.

## Author Contributions

**Conceptualization:** Chris Towlson, Sean Cumming, Jamie Salter, John Toner.

**Data curation:** Chris Towlson, Demi Jo Watson, John Toner.

**Formal analysis:** Chris Towlson, John Toner.

**Investigation:** Chris Towlson, Demi Jo Watson, Jamie Salter, John Toner.

**Methodology:** Chris Towlson, Jamie Salter, John Toner.

**Project administration:** Chris Towlson, Demi Jo Watson, John Toner.

**Supervision:** Chris Towlson, John Toner.

**Writing – original draft:** Chris Towlson, Sean Cumming, Jamie Salter, John Toner.

**Writing – review & editing:** Chris Towlson, Sean Cumming, John Toner.

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
