## [Decision Letter · Decision Letter 0]

25 Jul 2022

PONE-D-22-19101Soccer academy practitioners’ perceptions and application of bio-bandingPLOS ONE

Dear Dr. Towlson,

Thank you for submitting your manuscript to PLOS ONE. After careful consideration, we feel that it has merit but does not fully meet PLOS ONE’s publication criteria as it currently stands. Therefore, we invite you to submit a revised version of the manuscript that addresses the points raised during the review process.

We look forward to receiving your revised manuscript.

Kind regards,

Hugo Miguel Borges Sarmento

Academic Editor

PLOS ONE

Journal Requirements:

a) Did participants provide their written or verbal informed consent to participate in this study?

3. We note you have included a table to which you do not refer in the text of your manuscript. Please ensure that you refer to Table 3 in your text; if accepted, production will need this reference to link the reader to the Table.

Reviewers' comments:

Reviewer's Responses to Questions

**Comments to the Author**

1. Is the manuscript technically sound, and do the data support the conclusions?

Reviewer #1: Yes

Reviewer #2: Yes

2. Has the statistical analysis been performed appropriately and rigorously? 

Reviewer #1: N/A

Reviewer #2: Yes

3. Have the authors made all data underlying the findings in their manuscript fully available?

Reviewer #1: Yes

Reviewer #2: Yes

4. Is the manuscript presented in an intelligible fashion and written in standard English?

Reviewer #1: Yes

Reviewer #2: Yes

5. Review Comments to the Author

Reviewer #1: The paper about bio-banding summarizes results that have already been published using a questionnaire. Although I recognize the merits of this paper, there are some important points that should be clarified. The results were based on 25 participants using a questionnaire. The discussion is not easy to follow and needs to be adjusted.

[1] Line 75-79: Age at peak height velocity is an indicator of maturity timing and not status. Obviously, some studies use APHV to classify players as early, on time and late. However, the authors in the current study should not state that players who attained peak height velocity earlier are early maturing players. Briefly, maturity status is not equivalent to maturity timing. Please, adjust.

[2] Line 83. How do the authors assume that late maturing players are equally technically talented?

[3] The first paragraph is too long. Please, summarize the main aspects: sports selection. Two important papers should be included in the introduction. The second one uses a valid indicator of maturity status – skeletal age.

Konarski JM, Krzykała M, Skrzypczak M, et al. Characteristics of select and non-select U15 male soccer players. Biol Sport. 2021;38(4):535-544.

Malina RM, Peña Reyes ME, Eisenmann JC, Horta L, Rodrigues J, Miller R. Height, mass and skeletal maturity of elite Portuguese soccer players aged 11-16 years. J Sports Sci. 2000;18(9):685-693.

[4] Line 91: maturity offset is not APHV. References 9-11 are estimations of APHV.

[5] Line 95: Does maturity offset have been used to group players as pre-PHV, circa-PHV, post-PHV? Is this correct?

[6] Line 113: I do not agree with this sentence: “currently unclear to what extent bio-banding is being used, how it is being used, and for what purpose.”

The authors cited the paper on Sports Medicine (ref.23) – clearly, it explained the main issues related to bio-banding.

[7] Did the authors calculate the power sampling? Why 25 participants?

[8] Was the questionnaire validated?

[9] Line 612-615: Again, is not status. Why do the authors cite Koziel and Malina? The paper is not about a prediction.

[10] Line 623: I do not understand why the authors cited these references and stated “greater criterion biological maturity”. Towlson is a review and Salter only compared equations. Basically, none of these references tested the validity of equations. Please, adjust.

[11] The discussion is too long and not easy to follow, please summarize the main points.

Reviewer #2: The subject matter is relevant to readers of this journal, the rationale is consistent, and the approach is novel.

Also, the paper is very well written.

Therefore, I think that this manuscript represents a worthwhile and significant contribution to the body of literature. Please consider ta small detail; In the full title, short title and abstract authors use the term “soccer” while along the text authors reported to the main study object as “football”. Please homogenize.

6. PLOS authors have the option to publish the peer review history of their article (what does this mean?). If published, this will include your full peer review and any attached files.

Reviewer #1: No

Reviewer #2: No

---

## [Author Response · Author response to Decision Letter 0]

31 Jul 2022

Reviewer #1: The paper about bio-banding summarizes results that have already been published using a questionnaire. Although I recognize the merits of this paper, there are some important points that should be clarified. The results were based on 25 participants using a questionnaire. The discussion is not easy to follow and needs to be adjusted.

We thank reviewer 1 for the comments. We are particularly appreciative for their eye for detail. We feel the feedback given here was useful and improved the paper. 

[1] Line 75-79: Age at peak height velocity is an indicator of maturity timing and not status. Obviously, some studies use APHV to classify players as early, on time and late. However, the authors in the current study should not state that players who attained peak height velocity earlier are early maturing players. Briefly, maturity status is not equivalent to maturity timing. Please, adjust.

Thank you for this feedback and guidance. We completely agree, apologise for this oversight and have adjusted the statements according to reflect this feedback.

“The peak of accelerated growth in stature is typically referred to as peak height velocity (PHV) [10-13], and is often used as a growth landmark in which to assess maturity timing of adolescent soccer players [6, 14-17]. Using PHV as an identifiable reference point to assess player maturity timing is of relevance and importance to academy practitioners, given that players who undergo PHV earlier (i.e., achieve PHV earlier than peers) are often characterised as possessing temporary, enhanced stature, body-mass and key physical fitness (i.e., speed, power) characteristics are typically over-represented within academy programmes [3-5].”

[2] Line 83. How do the authors assume that late maturing players are equally technically talented?

Thank you for this comment. We do not assume later maturing players are equally talented.

“Such selection bias and playing behaviour are often to the detriment of later maturing players, who may well be equally technically talented, but are less likely afforded a fair playing environment in which to demonstrate their ability”

[3] The first paragraph is too long. Please, summarize the main aspects: sports selection. Two important papers should be included in the introduction. The second one uses a valid indicator of maturity status – skeletal age.

Konarski JM, Krzykała M, Skrzypczak M, et al. Characteristics of select and non-select U15 male soccer players. Biol Sport. 2021;38(4):535-544.

Malina RM, Peña Reyes ME, Eisenmann JC, Horta L, Rodrigues J, Miller R. Height, mass and skeletal maturity of elite Portuguese soccer players aged 11-16 years. J Sports Sci. 2000;18(9):685-693.

We have reduced the word count in the introduction and opening paragraphs. Thank you for alerting us to these papers. We have duly cited them.

[4] Line 91: maturity offset is not APHV. References 9-11 are estimations of APHV.

Thank you for raising this. We have adjusted the statement accordingly to reflect the cited references 

“Criterion18 measures for identifying maturation status are rarely available within applied sporting environments. Therefore, maturity offset (i.e., age at PHV – decimal age), age at PHV 10-12 or percentage of final adult height (PAH) 19 methods are often used to estimate maturation status 20.”

[5] Line 95: Does maturity offset have been used to group players as pre-PHV, circa-PHV, post-PHV? Is this correct?

Yes, this is correct. That said, we have added greater context to this statement.

“Bio-banding is the re-categorisation of adolescent athletes from chronological aged-ordered groupings into maturation specific groups (i.e., pre-PHV, circa-PHV or post-PHV or thresholds of percentage adult height24-31) with the assumed objective to reduce the large within-group variations ensued by individual variations in the timing of PHV across a specific age group.”

[6] Line 113: I do not agree with this sentence: “currently unclear to what extent bio-banding is being used, how it is being used, and for what purpose.”

The authors cited the paper on Sports Medicine (ref.23) – clearly, it explained the main issues related to bio-banding.

We do not fully agree with this point. The research conducted by Jamie Salter and colleagues does provide a nice understanding of how maturity status is measured and the how this is considered in relation to training load monitoring. However, this study does not exclusively address the extent to which bio-banding principles are applied in any reasonable detail. 

[7] Did the authors calculate the power sampling? Why 25 participants? 

No, we were unable to calculate power because we do not currently know how many practitioners are using bio-banding. The objective of this study was the aims were to examine the application of maturity status bio-banding within professional soccer academy programmes and understand the methods employed, the intended objectives, and the potential barriers to bio-banding. The extent in which bio-banding is being currently used was unknown prior to this study.

[8] Was the questionnaire validated?

The face and content validity of the survey was addressed via discussion with suitable academic staff (n = 3) who all possessed a relevant PhD and full-time academy soccer practitioners (n = 3) working within category 1 and 2 EPPP academy systems. as the items in the question here are being used for the purpose of information gathering, rather than the estimation of a specific construct, there was less need to consider are there aspects of psychometric integrity such as concurrent and predicted validity, structural validity, internal reliability or test retest reliability

[9] Line 612-615: Again, is not status. Why do the authors cite Koziel and Malina? The paper is not about a prediction.

Thank you. We have removed. 

[10] Line 623: I do not understand why the authors cited these references and stated “greater criterion biological maturity”. Towlson is a review and Salter only compared equations. Basically, none of these references tested the validity of equations. Please, adjust.

We in part agree. However, the review paper by Towlson et al provides an in-depth overview of many studies which what have addressed the validity and reliability of such methods. We understand that citing the original papers here may be considered intuitive. That said, the point being made in the current paper relates to a discussion point that is concurrently addressing the issue of validity etc with all of the equations which are used to bio-band players. Therefore, we feel its appropriate to initially direct the readers to the narrative presented in this paper. 

[11] The discussion is too long and not easy to follow, please summarize the main points.

Given the mixed-design of this study and the array of topics that bio-banding is associated with, we believe that the discussion in its current form is of an appropriate length. As this is the first study t explore the degree to which bio-banding is being used and perceived as effective, we also feel that is important that we take the opportunity to fully explore and communicate the practitioners perspectives. 

Reviewer #2: The subject matter is relevant to readers of this journal, the rationale is consistent, and the approach is novel.

Also, the paper is very well written.

Therefore, I think that this manuscript represents a worthwhile and significant contribution to the body of literature. Please consider ta small detail; In the full title, short title and abstract authors use the term “soccer” while along the text authors reported to the main study object as “football”. Please homogenize.

Thank you for the positive feedback. We have amended the title. 

---

## [Editor Report · Decision Letter 1]

23 Aug 2022

Soccer academy practitioners’ perceptions and application of bio-banding

PONE-D-22-19101R1

Dear Dr. Towlson,

We’re pleased to inform you that your manuscript has been judged scientifically suitable for publication and will be formally accepted for publication once it meets all outstanding technical requirements.

Kind regards,

Hugo Miguel Borges Sarmento

Academic Editor

PLOS ONE
---

## [Editor Report · Acceptance letter]

7 Dec 2022

PONE-D-22-19101R1 

Soccer academy practitioners’ perceptions and application of bio-banding 

Dear Dr. Towlson:

I'm pleased to inform you that your manuscript has been deemed suitable for publication in PLOS ONE. Congratulations! Your manuscript is now with our production department. 

Kind regards, 

on behalf of

Dr. Hugo Miguel Borges Sarmento 

Academic Editor

PLOS ONE